# OpenReview forum: "MR-GSM8K: A Meta-Reasoning Benchmark for Large Language Model Evaluation"
_ICLR.cc/2025/Conference — ICLR 2025 Poster_

### Official Review · Reviewer_PXiN · 2024-10-31

**Soundness:** 3
**Presentation:** 4
**Contribution:** 3
**Rating:** 6
**Confidence:** 5

**Summary:**

The paper proposes a novel paradigm for evaluating LLMs. Rather than assessing their ability to correctly produce a final answer, the paradigm assesses their ability to produce a more fine-grained analysis of given question-solutions paris. They focus on mathematical questions, specifically GSM8k. They generate a rather detailed dataset derived from GSM8k and include 3 types of base metrics for assessing the LLM in the evaluation process.

**Strengths:**

* The authors suggested paradigm shift to solution-driven evaluation (i.e., the correctness of the entire solution process) is indeed important, and not heavily studied
* The annotation process of the dataset with the three fields seems to be very useful in assessing fine-grained performance of LLMs
* Most claims and the steps taken by the authors in their research are fairly clear
* The process of training and selecting annotators
* They provide useful insights into LLM performance
  * Regarding “specialized”-trained models, specifically math models, they highlight their inability to generalize to related tasks
  * Larger models are not necessarily better than smaller ones
  * Fine-tuning LLMs on specific benchmarking tasks (such as GSM8k) may result in overfitting to the data in a detrimental way. Thus, fine-tuning should be handled with care
  * The impact of the number of correct solutions that is shown to the models during in-context-learning (i.e., “susceptibility of language models to the distribution of few-shot examples”)

**Weaknesses:**

* The authors explain the reasoning behind the 3 types of questions in their MR-GSM8k. Each type was previously suggested and studied: (1) sampling GSM8k, (2) modifying GSM8k to require code in the solution, (3) “reversed reasoning” from GSM8k. If these types were already studied, what is the novelty or added value in the suggested benchmark? This is unclear. Providing more information can help understand the novelty of their data generation process
* Discussion section
  * Subsection 6.1 is based on Appendix-D, and thus it is difficult to grasp the insights provided here. I suggest making this subsection clearer by providing some examples
* The authors claim that their suggested paradigm enables them to transform any benchmark. However, if I understand correctly, this is not really the case. First, they studied a math benchmark. In addition, the novelty here seems only marginal since their suggested benchmarking-transformation techniques (POT + reverse reasoning) was already suggested before. If there is indeed a claim for some novel method for transforming any benchmark, then it is not clear what the novel method is.

**Questions:**

* Table 1 - unclear what the values under the last column represent (First Error Steps). Is this the step where the first error occurred? Make it clearer
* Table 2
  * I suggest using explicit terms for task 1/2/3, or alternatively use explicit metrics where possible, such as ACC_{reason}, ACC_{step}. Currently, it is unclear. For example, what is the difference between “Task2-Accy” and “ACCstep” ?
  * please describe all abbreviation (TPR, TNR), including “k” in the table (they are not defined before the table is referenced)
* Lines 381 - 393: Point the reader to the few-shot examples that were given to the models (or did I miss it?)
* Conclusions section - the first paragraph is a summary of the paper, but does not contain conclusions

---

> ### Author Response · Authors · 2024-11-19
> **Response to reviewer PXiN (Part 1)**
>
> Dear Reviewer PXiN,
>
> Thank you for your detailed review and thoughtful feedback. We apologize for any confusion caused and hope our responses below address your concerns.
>
> **W1: Limited novelty of the dataset construction procedure**
>
> We would like to contend that **the primary contribution of our work lies in introducing the *Meta-Reasoning* evaluation paradigm** rather than the dataset construction itself. As noted in our paper (lines 110-113 and 461-467), our paradigm transforms traditional benchmarks into more robust and comprehensive assessment tools. By shifting from result-oriented question answering to process-oriented solution scoring, our approach redefines the evaluation landscape. Models must act as "solution-scoring teachers" rather than "QA students," a fundamental shift that requires more in-depth reasoning and scrutiny.
>
> Our paper demonstrates how this paradigm elevates benchmarks like GSM8K from being saturated and non-differentiating to becoming challenging tools that expose model weaknesses, especially concerning data contamination and memorization issues. This is evident in the struggles of several domain-finetuned expert models. Therefore, the novelty of our contribution lies in reframing the evaluation methodology, not just the dataset's problem structure.
>
> Additionally, our work exposes significant deficiencies **in the current training pipelines**. By conceptualizing step-by-step reasoning as Markov Decision Process trajectories within the solution space, we highlight the limitations of conventional supervised fine-tuning (SFT), which often relies on a narrow set of success trajectories. This lack of solution space coverage and contrastive reasoning hampers the development of robust, System-2-like thinking. Our findings underscore the need for more sophisticated training methods that incorporate reflection, self-correction, and counterfactual reasoning.
>
> To support our arguments, we hereby present the performance of the **o1 series models**, which utilize scaled computation budgets for reflection and backtracking:
>
> ### Experiment Results
>
> | **Model**                  | **Task 1 TPR (0-shot)** | **Task 1 TNR (0-shot)** | **Task 1 MCC (0-shot)** | **Task 2 Acc (0-shot)** | **Task 3 Acc (0-shot)** | **MR-Score (0-shot)** |
> |----------------------------|------------------------|------------------------|------------------------|------------------------|------------------------|-----------------------|
> | o1-mini-2024-09-12         | 93.3                  | 95.6                  | 89.0                   | 67.6                   | 62.2                   | 69.2                  |
>
> | **Model**                  | **Task 1 TPR (3-shot)** | **Task 1 TNR (3-shot)** | **Task 1 MCC (3-shot)** | **Task 2 Acc (3-shot)** | **Task 3 Acc (3-shot)** | **MR-Score (3-shot)** |
> |----------------------------|------------------------|------------------------|------------------------|------------------------|------------------------|-----------------------|
> | o1-mini-2024-09-12         | 93.3                  | 94.8                  | 88.1                   | 67.6                   | 61.8                   | 68.8                  |
>
> | **Model**                  | **Task 1 TPR (0-shot)** | **Task 1 TNR (0-shot)** | **Task 1 MCC (0-shot)** | **Task 2 Acc (0-shot)** | **Task 3 Acc (0-shot)** | **MR-Score (0-shot)** |
> |----------------------------|------------------------|------------------------|------------------------|------------------------|------------------------|-----------------------|
> | o1-preview-2024-9-12       | 89.3                  | 96.8                  | 86.6                   | 68.3                   | 65.7                   | 70.7                  |
>
> | **Model**                  | **Task 1 TPR (3-shot)** | **Task 1 TNR (3-shot)** | **Task 1 MCC (3-shot)** | **Task 2 Acc (3-shot)** | **Task 3 Acc (3-shot)** | **MR-Score (3-shot)** |
> |----------------------------|------------------------|------------------------|------------------------|------------------------|------------------------|-----------------------|
> | o1-preview-2024-9-12       | 84.4                  | 95.6                  | 80.8                   | 69.5                   | 66.6                   | 70.3                  |
>
> These results show that o1-mini and o1-preview models outperform previous state-of-the-art models like GPT-4 Turbo by a substantial margin, supporting our claim that models incorporating deliberate reasoning processes achieve superior performance.

---

> ### Author Response · Authors · 2024-11-19
> **Response to reviewer PXiN (Part 2)**
>
> **W2: Lack of in-context examples in Section 6.1**
>
> We apologize for the lack of clarity in Section 6.1. The reason that we summarized model behaviors instead of providing specific examples was originally due to space constraints. We have added these examples in the revised version of our paper we are uploading to the conference and appreciate your suggestion to enhance clarity. Below are the corresponding clarification.
>
> - The "reversal curse" refers to models failing to recognize that equations presented differently (e.g., 112 - 3x = 85 vs. 112 - 85 = 3x) are equivalent.
> - Errors involving units occur when models improperly combine quantities with different units, such as adding speed (5 km/h) to time (3 hours).
>
>
>
> **Clarifications on Tables 1 and 2 and the Conclusion Section**
>
> Thank you for pointing out the issues with the table captions and notations. We have also made the following clarifications in the revised version:
> - **Table 1**: The last two columns represent the average number of solution steps and the average step number where the first error occurs.
> - **Table 2**: We have renamed the columns to ACC_step, ACC_reason, and MCC for clarity, and provided definitions for abbreviations like TPR (True Positive Rate), TNR (True Negative Rate), and k (number of shots).
> - **Few-Shot Examples**: We have moved the few-shot examples from the supplementary materials into the appendix for easier reference.
> - **Conclusions**: We revised the conclusion to highlight our key findings and future directions rather than merely summarizing the paper.
>
> We hope these revisions and clarifications address your concerns. Thank you for your constructive feedback, and **we would be grateful if you could consider updating your assessment**. Wishing you a wonderful day!

---

### Official Review · Reviewer_HcPg · 2024-11-02

**Soundness:** 2
**Presentation:** 2
**Contribution:** 3
**Rating:** 3
**Confidence:** 4

**Summary:**

The authors to propose a new evaluation paradigm that shifts the role of models from “QA student” to a “solution-scoring teacher.” In this approach, instead of generating solutions to questions, models are presented with question-solution pairs and are have to determine the correctness of each solution, identifying the first error step if any, and providing explanations for errors. This paradigm encourages models to engage in meta-reasoning, assessing different reasoning processes rather than simply arriving at correct answers. The authors developed a meta-reasoning version of the GSM8K benchmark, called MR-GSM8K, and introduced a new metric, the MR-Score, which is a weighted combination of three submetrics. Instances in MR-GSM8K are manually labeled by experts to ensure accuracy in evaluation. Their findings indicate that specialized math models struggle to generalize to this new benchmark, and, contrary to expectations, larger models do not consistently perform better in tasks on MR-GSM8K. The authors argue that models frequently exhibit superficial reasoning in math tasks and link these to their possible limitations in “System 2” thinking—i.e., that models fail to engage in slow, deliberate reasoning that examines assumptions, conditions, and logic for thorough error detection.

**Strengths:**

I commend authors' general aims in evaluation of LLMs' math reasoning steps rather than just final solutions as well as their efforts to ensure minimise the annotation errors and quality control the benchmark instances. I also like the metric MR-Score they introduced to evaluate model performance. Introducing a meta math reasoning benchmark is also important in advancing our understanding of LLMs abilities.

**Weaknesses:**

I find it difficult to understand why, given their goal of assessing LLMs’ step-by-step math reasoning abilities, the authors chose to create a meta-reasoning benchmark where the task is to evaluate the correctness of provided step-by-step math solutions rather than to generate correct step-by-step reasoning. Evaluating solutions is a distinctly different skill from producing them. Just as we wouldn’t expect students who excel at solving math problems to also excel at verifying others' solutions, we shouldn’t assume these skills are interchangeable for models.

Furthermore, the authors' results might even suggest that their benchmark tests different capabilities than those measured by GSM8K. For instance, despite the claims in lines 375-379, the results in Table 2 indicate that none of the models specialized in solving math problems performed well on their benchmark. In lines 377-378, the authors claim that MAmmoTH-70B and DeepSeek-Math-7B-RL achieve decent performance. However, Table 2 shows that their MR-Scores are near zero in multi-shot learning and only slightly higher in zero-shot learning. If Table 2 is accurate, this could indicate that their benchmark does not truly measure the same abilities as GSM8K, MATH, or other similar benchmarks.

Given this, it’s unclear how the authors can argue that fine-tuning on math problems like those in GSM8K leads to overfitting or only a superficial mastery of mathematical reasoning, especially if their benchmark assesses a different skill than GSM8K and similar tests. Why should we expect high performance on GSM8K to correlate with high performance on MR-GSM8K?

The authors need to clarify their definition of "mathematical reasoning." How does their concept of math reasoning differ (or not) from the concept tested by benchmarks like GSM8K? Are these two concepts comparable, or do they address entirely separate abilities?

**Questions:**

If the authors are interested in evaluating LLMs' math reasoning rather than just their ability to produce a correct answer, why not have the models generate final answers using Chain-of-Thought (CoT) or Program-of-Thought (PoT) reasoning and evaluate those directly, rather than assessing the models' meta-reasoning abilities?

What do the authors mean by "mathematical reasoning"? Are they evaluating the same abilities as those tested by GSM8K and similar benchmarks?

---

> ### Author Response · Authors · 2024-11-19
> **Response to reviewer HcPg (Part 1)**
>
> Dear Reviewer HcPg,
>
> Thank you for your detailed review and thoughtful questions. We are glad to address your concerns below:
>
> **Q1: Is evaluating a solution different from producing one?**
>
> This is a very insightful question. We argue that evaluating solutions and producing them are mostly **equivalent skills** for several reasons.
>
> Firstly, mathematical word problems have long been used to assess the reasoning abilities of language models. When we test LLMs on math problems, we are essentially evaluating their ability to recognize patterns, understand relationships between objects, correctly apply formulas, and logically combine conditions and assumptions. Traditionally, metrics have focused on the final computation result as a proxy for reasoning ability, mainly due to implementation convenience and lack of annotated process oriented data and corresponding evaluation methods. However, this result oriented approach overlooks the accuracy of intermediate steps, which are crucial in assessing the quality of the reasoning process.
>
> **Our meta-reasoning paradigm addresses this gap**. When models are asked to score the correctness of solutions, they engage in the same fundamental skills: discerning patterns/relations/conditions, performing counterfactual reasoning that includes similar applications of formulas and computation but with an extra step of contrastive comparison. Thus, our solution-scoring benchmark **serves the same overarching goal of assessing robust reasoning abilities** but does so in a more holistic, challenging and process-oriented manner.
>
> To support our claim, we present the results of the o1-mini and o1-preview models. These models reportedly combine self-reflection and self-correction processes (similar to meta-reasoning paradigm) into their reasoning procedures. They exhibited substantial improved performance on our meta-reasoning benchmark—something that is hard to observe with GSM8K due to the saturation and lack of differentiation.
>
> ### Experiment Results
>
> | **Model**                  | **Task 1 TPR (0-shot)** | **Task 1 TNR (0-shot)** | **Task 1 MCC (0-shot)** | **Task 2 Acc (0-shot)** | **Task 3 Acc (0-shot)** | **MR-Score (0-shot)** |
> |----------------------------|------------------------|------------------------|------------------------|------------------------|------------------------|-----------------------|
> | o1-mini-2024-09-12         | 93.3                  | 95.6                  | 89.0                   | 67.6                   | 62.2                   | 69.2                  |
>
> | **Model**                  | **Task 1 TPR (3-shot)** | **Task 1 TNR (3-shot)** | **Task 1 MCC (3-shot)** | **Task 2 Acc (3-shot)** | **Task 3 Acc (3-shot)** | **MR-Score (3-shot)** |
> |----------------------------|------------------------|------------------------|------------------------|------------------------|------------------------|-----------------------|
> | o1-mini-2024-09-12         | 93.3                  | 94.8                  | 88.1                   | 67.6                   | 61.8                   | 68.8                  |
>
> | **Model**                  | **Task 1 TPR (0-shot)** | **Task 1 TNR (0-shot)** | **Task 1 MCC (0-shot)** | **Task 2 Acc (0-shot)** | **Task 3 Acc (0-shot)** | **MR-Score (0-shot)** |
> |----------------------------|------------------------|------------------------|------------------------|------------------------|------------------------|-----------------------|
> | o1-preview-2024-9-12   | 89.3                  | 96.8                  | 86.6                   | 68.3                   | 65.7                   | 70.7                  |
>
> | **Model**                  | **Task 1 TPR (3-shot)** | **Task 1 TNR (3-shot)** | **Task 1 MCC (3-shot)** | **Task 2 Acc (3-shot)** | **Task 3 Acc (3-shot)** | **MR-Score (3-shot)** |
> |----------------------------|------------------------|------------------------|------------------------|------------------------|------------------------|-----------------------|
> | o1-preview-2024-9-12   | 84.4                  | 95.6                  | 80.8                   | 69.5                   | 66.6                   | 70.3                  |
>
>
> These results illustrate that models with effective self-reflection, self-correction, and deliberate reasoning (e.g. the same set of skills Meta-Reasoning is evaluating) excel on our benchmark (with an **absolute improvement of 20%** against GPT4-Turbo). This suggests that the ability to scrutinize and reason through each step is indeed a fundamental component of robust reasoning.

---

> ### Author Response · Authors · 2024-11-19
> **Response to reviewer HcPg (Part 2)**
>
> **Q2: Do the results of math-specialized models contradict GSM8K results?**
>
> We understand your concern and wish to clarify that the poor performance of specialized math models on MR-GSM8K does not imply that our benchmark measures an entirely different set of abilities. Rather, it highlights the potential overfitting to specific question types and formats. Specifically:
>
> 1. **Format Overfitting**: Specialized models, even with few-shot demonstrations, exhibited overfitting to the GSM8K response format. For example, they often used specific format indicators like "###" and tended to provide question-answering outputs, deviating from the expected solution-scoring behavior.
> 2. **Bias Towards Correctness**: These models frequently (>90% of cases) predicted solutions as "correct," regardless of the ground truth, revealing a lack of genuine evaluative reasoning.
> 3. **Degraded Reasoning**: Their explanations for errors were often nonsensical or incorrect, further indicating superficial reasoning.
>
> Our observations align with Goodhart's Law: optimizing a specific measure (e.g., GSM8K scores) may lead to overfitting, rendering the measure ineffective at capturing true reasoning abilities. While fine-tuning on correct solutions may boost performance on traditional benchmarks, it does not necessarily foster deeper reasoning skills. As elaborated in Section 6 and illustrated by Figure-3 of our paper, we believe our paradigm indicates that a more fundamental way to enhance reasoning is to provide in depth analysis/contrast on each step of the solutions. Our hypothesis is verified from the success of the o1 series models, which, through reflective and iterative thinking, demonstrate superior performance and generalization across a broader domain.
>
> We hope these clarifications address your concerns and illustrate the value of our meta-reasoning approach. We truly appreciate your kind review and if you find the above addresses your concerns, **we would be truly grateful if you can consider adjusting your final rating for us**. We wish you a wonderful day ahead. Thanks!

---

> ### Author Response · Authors · 2024-11-23
> **Request for Discussion and Feedback on Rebuttal Responses**
>
> Dear Reviewer HcPg,
>
> Thank you for your detailed review and valuable feedback. We have carefully addressed your concerns in our rebuttal and would greatly appreciate your engagement during the rebuttal period to discuss whether our responses sufficiently address them. Your insights are highly valuable, and we are happy to provide further clarification if needed.
>
> Thank you again for your time and thoughtful review.
>
> Best regards,

---

> > ### Comment · Reviewer_HcPg · 2024-11-24
> >
> > I thank the authors for their comments. However, I strongly disagree that evaluating a solution and producing a solution are equivalent skills, as the authors are arguing. To generate a solution, it is not enough just to identify patterns and spot errors or determine whether formulas have been correctly applied. What is needed for generation is, for example, applying knowledge and heuristics to navigate through the problem, planning and strategizing to get to a solution, and combining various concepts and methods to construct a solution. None of this is needed for solution evaluation. The authors also present a table to support the claim that, I guess, evaluating a solution and producing it are equivalent skills. However, all that the table is showing is that GSM8K is saturated and the benchmark they've created isn't. This doesn't provide any support for the claim that evaluation and production are equivalent skills.
> >
> > Furthermore, the second part of the authors' answer doesn't really address my concern that what they show in Table 2 and what they say in lines 375–379 and 377–378 of their paper doesn't really add up. Again, the authors claim that 'Deepseek-Math-7B-RL and MAmmoTH-70B managed to comprehend our complex evaluation instructions and achieved decent performance.' Looking at Table 2, we find that both models performed poorly, with performance close to 0 as measured by MR-score when k=3 and only slightly better when k=1. The reason why I was saying that this might indicate that they are doing something different from what MR-GSM8K wants them to is exactly because of the reasons the authors provide in their Part 2 response. These models are fine-tuned to generate solutions to the problems, not to evaluate them. Given this, it is then expected that these models won't adhere to the different format required by MR-GSM8K; it is also expected that they provide nonsensical explanations for their evaluation and lack evaluative reasoning because they are not built to evaluate solutions but to produce them. This, to me, further indicates that evaluation and solution production are completely different and one cannot consider them equivalent. For this reason, evaluating models that are specialized for solution production on problems that are about evaluating solutions is not appropriate.
> >
> > Given all this, I am inclined not to revise my initial assessment.

---

> > > ### Author Response · Authors · 2024-11-25
> > > **Response to reviewer HcPg**
> > >
> > > Dear Reviewer HcPg,
> > >
> > > Thank you for your thoughtful feedback and detailed reply. We acknowledge that our use of the term *equivalent* in describing solution scoring and solution generation may have been too strong. While scoring a solution often requires solving or partially solving a problem (e.g., identifying the first error step), we agree that solution generation involves additional tasks.
> > >
> > > That said, we believe that comparing solution scoring and solution generation abilities is valuable for several reasons, as outlined in our paper:
> > >
> > > 1. **Real-world Applicability**: In scenarios such as consultancy and teaching, the ability to explain the rationale behind solutions and effectively contrast different approaches is crucial for explainability and trustworthiness.
> > >
> > > 2. **Revealing Deficiencies**: Comparing these two abilities helps uncover fundamental issues such as reversal curses and lack of ontological understanding. These shortcomings highlight gaps in reasoning and generalization capabilities.
> > >
> > > 3. **Enabling Self-Correction**: A lack of scoring ability hinders models from conducting effective self-correction and iterative improvements. Conversely, models with strong scoring abilities, such as the o1 series (as demonstrated in our benchmarks), exhibit greater robustness and reasoning power.
> > >
> > > We appreciate your concerns regarding the interpretation of Table 2 and will revise our phrasing in the updated version of the paper to better reflect the nuances of our findings. Specifically, we will clarify that while models like DeepSeek-Math-7B-RL and MAmmoTH-70B are fine-tuned for solution generation, their poor performance on MR-GSM8K highlights the distinction between generation and evaluation tasks, supporting our argument that more holistic reasoning abilities are required for robust evaluation.
> > >
> > > Your feedback has been invaluable, and we are incorporating it into the revised version of the paper being uploaded to the conference. We believe our paradigm offers a new perspective on evaluating reasoning skills, exposing critical deficiencies in current models, and suggesting potential avenues for improvement, such as integrating System-2 thinking.
> > >
> > > Please let us know if the above response addresses your concerns. Thank you again for your valuable input.
> > >
> > > Best regards,

---

### Official Review · Reviewer_wbNK · 2024-11-03

**Soundness:** 4
**Presentation:** 4
**Contribution:** 4
**Rating:** 8
**Confidence:** 4

**Summary:**

This paper proposes a new benchmark category, Meta-Reasoning (reasoning about reasoning), and a new corresponding meta-reasoning benchmark dataset, MR-GSM8K, derived from the well-known GSM8K. In this benchmark, models are shown question-solution pairs and are tasked with identifying correct solutions, incorrect steps, and explaining incorrect steps. All three of these subtasks are captured in their proposed novel metric, the MR-score. They show that their benchmark indeed tests for a new capability and is quite challenging - models that perform well on the reasoning dataset GSM8K surprisingly perform substantially worse on MR-GSM8K.

**Strengths:**

Originality
-The benchmark task and evaluation metric appears to be novel. It is testing for a new capability, reasoning about reasoning.

Quality
-Their benchmark has led to strong insights that may influence future research. For example, they show that bigger models are not necessarily better than smaller models on their challenging meta-reasoning benchmark, with Phi-3 beating models many times larger.

-The creation of the benchmark dataset underwent several rounds of reviews from LLMs and human annotators.

Clarity
-The paper is logically organized and easy to follow.

Significance
-It seems that multi-step reasoning datasets don’t fully assess the steps taken to arrive at a solution; instead, they score a model based on its final answer. So there is a need for more nuanced evaluation, such as this work. This work challenges the surface-level assessments of previous reasoning benchmarks and will inspire deeper research into enhancing model reasoning.

**Weaknesses:**

-Table 2 could be easier to read, see suggestions below.

**Questions:**

-Table 2 needs more clarity: consider adding a table description defining your abbreviations “Task1-TPR”, etc. Also the values that are bolded were not intuitive - in the table description also explain what bold means. Also explain what k=0, k=3 means.

---

> ### Author Response · Authors · 2024-11-19
> **Response to reviewer wbNK**
>
> Dear Reviewer wbNK,
>
> Thank you for your thoughtful and encouraging feedback. We are delighted to address your concerns and appreciate the opportunity to improve our work.
>
> **Clarification on Table 2**
>
> We apologize for any confusion caused by the presentation of Table 2. The bolded values indicate the best-performing models within their respective categories. Regarding the abbreviations TPR (True Positive Rate) and TNR (True Negative Rate), we explained them briefly in the main text (line 354) when introducing Table 2 in Section 5.2: “Our evaluation results are shown in Table 2... For Task 1, we also report the true positive rate and true negative rate.” Similarly, the k=0 and k=3 annotations refer to zero-shot and few-shot settings, respectively, as mentioned in Section 5.1 (line 323).
>
> We understand that these explanations were not sufficiently clear in the table caption. **We are uploading a new version of our paper** and we have explicitly defined these terms in the table description and ensure the caption is more intuitive and reader-friendly.
>
> **Thank you once again for your valuable feedback. We hope this clarifies your concerns, and we wish you a wonderful day**!

---

### Official Review · Reviewer_AEiU · 2024-11-03

**Soundness:** 3
**Presentation:** 3
**Contribution:** 2
**Rating:** 6
**Confidence:** 4

**Summary:**

This paper proposes MR-GSM8k a new evaluation dataset adopted from GSM8k to test the depths of cognitive abilities in LLMs. This dataset consists of samples from GSM8k, code based solutions to the problems, and reverse-engineered versions asking the model to find the value of an input variable given the answer. Rather than relying on surface-level final answer match, they aim to score the model for its reasoning ability by proposing a new metric MR score - which given a <task, solution> pair, expects the model to evaluate its correctness, identify the first point of error, and the reason for the error. Evaluating a range of models, they demonstrate that these models perform similarly on the original GSM8k but show wide variance on MR-GSM8k as it tests their core ability to reason, and investigate possible reasons for this, urging for benchmarks that go deeper than surface-level evaluations.

**Strengths:**

- The construction of this new dataset is quite novel and opens doors to more nuanced evaluation methods to test the reasoning of language models.
- With increasing interest in the interpretability of LMs, this work transforms an existing dataset to measure which parts models fail.
- They highlight the memorization issue prevalent in LMs — it has imitation skills but lacks a deep understanding of the underlying logical as they can solve the task but not infer with the same accuracy if a given solution is correct or not.

**Weaknesses:**

- The dataset construction procedure is of limited novelty with only the reversal mode being significantly different.
- A more detailed error analysis with some discussion on actionable insights on how to combat them would be beneficial to the community
Several works have already probed the memorization effect and lack of genuine reasoning in LLMs on different tasks (Embers of Autoregression, McCoy et. al; Deciphering the Factors Influencing the Efficacy of Chain-of-Thought, Prabhakar et. al; GSM-Symbolic, Mirzadeh et al.) providing concrete insights into the error patterns.
- Given these weaknesses, I feel the dataset could be of interest to a section of the community.

**Questions:**

N/A

---

> ### Author Response · Authors · 2024-11-19
> **Response to reviewer AEiU (part 1)**
>
> Dear Reviewer AEiU,
>
> Thank you for your insightful review and constructive feedback. We hope our response below addresses your concerns adequately.
>
> **W1: Limited novelty of the dataset construction procedure**
>
> We acknowledge your observation and would like to clarify that our primary contribution lies not merely in the dataset itself but **more** in introducing the *Meta-Reasoning* evaluation paradigm. As highlighted in our paper (lines 110-113 and 461-467), this paradigm enables transforming any existing reasoning benchmark into a more robust and comprehensive assessment tool. Specifically, our approach shifts the evaluation focus from result-oriented question answering to process-oriented solution scoring. This transition is akin to changing roles from a student, who merely answers questions, to a teacher, who critically assesses the correctness and reasoning process behind a solution. Consequently, the evaluation poses a more substantial challenge to models.
>
> We devoted significant portions of our paper to illustrating (1) how our paradigm transforms a benchmark like GSM8K from a non-differentiating, saturated metric into one that effectively distinguishes models, and (2) how this transformation enhances robustness against data contamination and memorization, which are common issues arising from opaque pretraining and fine-tuning processes. For instance, our results demonstrate how several domain-finetuned expert models struggled with our benchmark, underscoring the paradigm's strength. Thus, **the framing of problems, while important, is secondary to our primary contribution: introducing a more comprehensive and effective meta-reasoning assessment methodology**.

---

> ### Author Response · Authors · 2024-11-19
> **Response to reviewer AEiU (part 2)**
>
> **W2: Discussion on actionable insights**
>
> We appreciate your suggestion and agree that a discussion on actionable insights would benefit our work. Accordingly, we have conducted additional experiments, and below, we outline some key insights and potential implications.
>
> As discussed in Section 6.2 and depicted in Figure 3, one of the core contributions of our meta-reasoning paradigm is **uncovering critical weaknesses in both the evaluation and training pipelines** of current models. Viewing the step-by-step reasoning of various questions as Markov Decision Process (MDP) trajectories within the solution space reveals that most supervised fine-tuning (SFT) approaches focus on a limited number of successful trajectories. This results in incomplete coverage of the solution space, lacking contrastive understanding among different solutions, discernment of assumptions, or counterfactual reasoning—all vital elements of *System-2* thinking, necessary for robust, reflective reasoning.
>
> To support our hypothesis, we evaluated our MR-GSM8K benchmark on the newly developed **o1 series of models**, which utilize scaled computational budgets at test time to engage in self-reflection, self-correction, backtracking, and counterfactual reasoning. These models mimic System-2 thinking, demonstrating broader knowledge and coverage of the solution space. As anticipated, models with these capabilities performed notably well in our benchmark, corroborating our hypothesis.
>
> ### Experiment Results
>
> | **Model**                  | **Task 1 TPR (0-shot)** | **Task 1 TNR (0-shot)** | **Task 1 MCC (0-shot)** | **Task 2 Acc (0-shot)** | **Task 3 Acc (0-shot)** | **MR-Score (0-shot)** |
> |----------------------------|------------------------|------------------------|------------------------|------------------------|------------------------|-----------------------|
> | o1-mini-2024-09-12         | 93.3                  | 95.6                  | 89.0                   | 67.6                   | 62.2                   | 69.2                  |
>
> | **Model**                  | **Task 1 TPR (3-shot)** | **Task 1 TNR (3-shot)** | **Task 1 MCC (3-shot)** | **Task 2 Acc (3-shot)** | **Task 3 Acc (3-shot)** | **MR-Score (3-shot)** |
> |----------------------------|------------------------|------------------------|------------------------|------------------------|------------------------|-----------------------|
> | o1-mini-2024-09-12         | 93.3                  | 94.8                  | 88.1                   | 67.6                   | 61.8                   | 68.8                  |
>
> | **Model**                  | **Task 1 TPR (0-shot)** | **Task 1 TNR (0-shot)** | **Task 1 MCC (0-shot)** | **Task 2 Acc (0-shot)** | **Task 3 Acc (0-shot)** | **MR-Score (0-shot)** |
> |----------------------------|------------------------|------------------------|------------------------|------------------------|------------------------|-----------------------|
> | o1-preview-2024-9-12   | 89.3                  | 96.8                  | 86.6                   | 68.3                   | 65.7                   | 70.7                  |
>
> | **Model**                  | **Task 1 TPR (3-shot)** | **Task 1 TNR (3-shot)** | **Task 1 MCC (3-shot)** | **Task 2 Acc (3-shot)** | **Task 3 Acc (3-shot)** | **MR-Score (3-shot)** |
> |----------------------------|------------------------|------------------------|------------------------|------------------------|------------------------|-----------------------|
> | o1-preview-2024-9-12   | 84.4                  | 95.6                  | 80.8                   | 69.5                   | 66.6                   | 70.3                  |
>
> The o1-mini and o1-preview models demonstrate significant improvements across all metrics compared to GPT4-Turbo, with an **absolute performance increase of around 20%**. This supports our assertion that incorporating a slow, deliberate thinking paradigm enables more robust reasoning capabilities in LLMs. While our current results are promising, we believe a more thorough investigation into scaling computational budgets at train/test time is beyond the scope of our paper and we would like to leave that in our future work.
>
> We truly appreciate your kind review and if you find the above addresses your concerns, **we would be truly grateful if you can consider adjusting your final rating for us**. We wish you a wonderful day ahead. Thanks!

---

> > ### Comment · Reviewer_AEiU · 2024-11-24
> >
> > I thank the authors for their clarifications, and would like to keep my initial score.

---

### Meta-Review · Area_Chair_9oGK · 2024-12-20

**Metareview:**

The paper is interested in shifting the task of an LLM, from producing a solution to a query, to evaluating a pair (query, answer): scoring its correctness and possibly identifying where there is a failure and why. The metaphor is that of shifting the LLM from a system-1 to a system-2 level.
Experimentally, the performance indicators based on the considered tasks (scoring, failure identification) appear to be more informative (finer-grained than the performance) w.r.t. the considered LLMs.

**Additional Comments On Reviewer Discussion:**

A most interesting part of the discussion is whether evaluating solutions and producing them are mostly equivalent skills. The authors say yes (but the paragraph below says the contrary: "[the] result oriented approach [does not necessarily enable to judge of] the quality of the reasoning process").
Reviewer HcPg and the area chair hold a different opinion.
All in all, measuring the ability to judge the quality of a reasoning process appears a precious skill, and I think the paper makes a true contribution here.
If the new score MR-GSM8K could be predictive of the performance system when applied to "out of training distribution" cases, then the paper would make a major contribution.

---

### Decision · Program_Chairs · 2025-01-22

Accept (Poster)